# Deep Learning Modeling of Androgen Receptor Responses to Prostate Cancer Therapies

**DOI:** 10.3390/ijms21165847

**Published:** 2020-08-14

**Authors:** Oliver Snow, Nada Lallous, Martin Ester, Artem Cherkasov

**Affiliations:** 1School of Computing Science, Simon Fraser University, Burnaby, BC V5A 1S6, Canada; oliver_snow@sfu.ca (O.S.); ester@sfu.ca (M.E.); 2Vancouver Prostate Centre, University of British Columbia, 2660 Oak St, Vancouver, BC V6H 3Z6, Canada; nlallous@prostatecentre.com

**Keywords:** prostate cancer, androgen receptor, deep learning, proteochemometrics

## Abstract

Gain-of-function mutations in human androgen receptor (AR) are among the major causes of drug resistance in prostate cancer (PCa). Identifying mutations that cause resistant phenotype is of critical importance for guiding treatment protocols, as well as for designing drugs that do not elicit adverse responses. However, experimental characterization of these mutations is time consuming and costly; thus, predictive models are needed to anticipate resistant mutations and to guide the drug discovery process. In this work, we leverage experimental data collected on 68 AR mutants, either observed in the clinic or described in the literature, to train a deep neural network (DNN) that predicts the response of these mutants to currently used and experimental anti-androgens and testosterone. We demonstrate that the use of this DNN, with general 2D descriptors, provides a more accurate prediction of the biological outcome (inhibition, activation, no-response, mixed-response) in AR mutant-drug pairs compared to other machine learning approaches. Finally, the developed approach was used to make predictions of AR mutant response to the latest AR inhibitor darolutamide, which were then validated by in-vitro experiments.

## 1. Introduction

Resistance to drug treatments is a common occurrence across many diseases, but it is especially prevalent and lethal in cancers, where it represents a major obstacle for long-term therapies. The acquired drug resistance can be caused by a number of different mechanisms by which cancer cells can escape the treatment [1]. One of the major ways is the development of gain-of-function mutations where the target protein becomes altered under selective therapeutic pressure, rendering the drug ineffective. Gain of function mutations are of particular importance in prostate cancer (PCa), where resistance is a common and often deadly occurrence [2].

The main drug target in PCa is the androgen receptor (AR), a nuclear hormone receptor whose increased activation is one of the principal drivers of PCa. Multiple decades of research on the AR has led to a number of targeted drug treatments that have significantly improved patient survival and well-being. However, despite the major gains in AR targeted treatments, resistance invariably develops to all current drugs [3,4]. One important aspect of resistant AR mutants is that they do not simply render the drug ineffective but can even turn the drug from an antagonist into an agonist, thus promoting cancer growth. This characteristic seems to be unique to the AR and thus emphasizes the need to identify gain-of-function mutations that cause this phenotype so that patients can be screened and taken off treatment before resistance develops [5]. Additionally, understanding and predicting those mutations that cause resistance will enable us to design better drugs that might avoid this mechanism in the future.

Previous research had identified a number of new AR mutants from circulating cell-free DNA of PCa patients using next generation sequencing technology [6]. The inhibition of these mutants (along with others documented in the literature) were then experimentally measured in PCa cell lines in response to increasing doses of the current and upcoming AR inhibitors. Building upon the previous work, additional mutants listed in the McGill Androgen Receptor Gene Mutations Database [7] were screened against first and second generation anti-androgens, resulting in a high quality dataset of 68 distinct AR mutants observed in PCa patients and described in the literature.

Naman et al. [8], in 2016, introduced QSAR (quantitative structure-activity relationship) modeling as a way of binary classification of the AR mutant responses to common PCa drugs and endogenous human steroids using structure-based 4D QSAR descriptors [9,10,11,12]. The QSAR model was able to identify a de novo AR mutant that demonstrated a resistant phenotype to the current anti-androgen drugs. However, the approach was relatively simplistic, and, despite achieving an accuracy of ~80%, the model generated several false positive predictions for the external validation set. Importantly, the use of binary antagonist/agonist classification reduced the complexity of the AR mutant responses that can range from full antagonism to full agonism and also includes partial agonistic response and no-response (non-functional AR mutants) categories.

Recent advances in machine learning theory have led to significant progress and qualitative changes in QSAR and chemoinformatics practice. Interpretable linear QSAR models have become increasingly replaced by support vector machines (SVM), random forest (RF), artificial neural networks (ANN), and other non-linear techniques that demonstrate robust performance on large biological datasets [13,14]. More recently, the progress in deep learning techniques along with the increasing availability of biological data has brought deep neural networks (DNNs) into the drug modeling spotlight [15]. DNNs have been particularly effective due to their ability to learn more complex non-linear trends from larger datasets and their lower requirements for input representations, i.e., lesser need for precise descriptor engineering [16,17].

In this work, we utilized a DNN, along with general proteochemometric descriptors, to predict more differentiated AR mutant-drug responses on a significantly extended experimental dataset, compared to our previous study. Furthermore, we implemented a structure-independent protocol where, by using protein sequence-based descriptors and 2D drug fingerprints, we avoided the drug docking step in feature construction, saving time and making the model more generalizable. Notably, such an approximation allows the consideration of mutations that occur outside of the ligand-binding domain (LBD) and DNA binding domain (DBD) of the AR, where no structural information is available. As mentioned above, the resulting DNN model distinguishes four AR mutant response phenotypes and provides rather accurate discrimination between them.

## 2. Results and Discussion

### 2.1. Training Dataset

AR mutant data for training was collected as described by Lallous et al. [6] in which single and multiple amino acid substitutions of the AR were expressed in PC3 cells. Responses to seven different anti-androgens (bicalutamide [18], enzalutamide [19,20], hydroxyflutamide [21], Apalutamide [22], darolutamide [23], and 2 in-house developed AR inhibitors, VPC-13566 and VPC-13789) were measured in a luciferase-reporter assay. Chemical structures of experimental antiandrogens can be seen in Appendix A. An example of the measured responses can be seen in Figure 1 where 3 of the main phenotypes (agonist, mixed response, and antagonist) are displayed for the mutant W742C. The dose-response curve data consisting of 11 increasing drug concentrations (see Section 4.1) and their transcriptional activity values (normalized to wild-type) were converted to 4 classes corresponding to the above phenotypes with an additional non-responsive phenotype (see Section 4.2 for details on how curves were converted to classes). The dataset was split into a training set (80% of initial dataset), upon which cross validation and hyperparameter tuning was performed, and a test set (20% of initial dataset) to evaluate the performance of the methods. The split was performed in a stratified manner so that all compounds were represented in both the train and test sets.

### 2.2. Feature Calculation

Using the fasta-formatted amino acid sequences of the AR variants (trimmed to only include the range in which the current substitutions occur), mutant sequences were generated for each single or multiple mutant in the dataset. These sequences were then used to generate z-scale descriptors using the *iLearn* python package [24]. Z-scale descriptors characterize each amino acid by 5 physico-chemical values and have been shown to consistently match or outperform other sequence based descriptors on benchmark datasets [25,26]. Drug molecules were encoded with extended connectivity fingerprint (ECFP; also known as Morgan fingerprint) descriptors using the RDKit Python package and a radius of 2, resulting in a 2048 bit vector for each molecule [27,28]. Z-scale descriptors were then standardized to zero-mean and unit variance, while Morgan fingerprints were left as one-hot encoded vectors.

Given that certain drug response phenotypes are less frequent than others (resistant phenotype is relatively rare), there is a high imbalance between classes that can negatively affect the accuracy of the model (the model can get high accuracy by just predicting the majority class). To address this imbalance, we over-sample the minority classes according to the borderline synthetic minority over-sampling technique (SMOTE) method, which outperformed the regular SMOTE method and random oversampling in cross validation [29,30].

### 2.3. Baseline Methods

To justify the use of the more complex DNN method for this task, we benchmarked the corresponding results against outcomes from SVM and RF models. In particular, we obtained 5-fold cross validation predictions by SVM and RF and utilized area under the receiver operating characteristic curve (AUC) statistics. Hyperparameter tuning was performed using a random search 5-fold cross-validation on a dictionary of a range of parameters. The parameters that gave the best validation accuracy were chosen. SVM used an radial basis function kernel with gamma=0.001, and the RF was built with # of estimators=50 and max depth=10. Oversampling was performed within the cross validation loop so as not to bleed any information into the validation set.

### 2.4. DNN Model Training

The general overview of our approach can be seen in Figure 2, where the network takes in protein sequence and 2D chemical descriptors and outputs 4 classes of dose-response curve. The DNN was also optimized to find the best hyperparameters using 5-fold cross validation. The optimal architecture was found to be two hidden layers with 128 neurons in the first layer and 32 in the second layer, which is a relatively small network but is a suitable amount of weights to train given the small number of training examples. Categorical cross-entropy loss was used, with a batch size of 16 and the ADAM (adaptive moment estimation) optimizer for training [31]. Each hidden node had a rectified linear unit (ReLU) [32] activation applied, and the output layer had a softmax activation applied [33]. Additionally, to reduce over-fitting of the network, a dropout rate of 0.01 was applied to each hidden layer and early stopping was used to terminate training if validation loss did not improve over 20 epochs.

### 2.5. Prediction Results

Interestingly, SVM and random forest have fairly high AUC on the training set, despite the large dimension of the input, and the DNN performs only slightly better than SVM and the same as random forest (Figure 3). However, further investigation into precision and recall shows that both baseline methods over-predict for the majority class, a bias that the AUC does not fully capture. This can be seen in Table 1 where the DNN significantly outperforms the baselines on the test set in regard to precision, recall, and F1 score, in both per-class performance and in the average of each metric, weighted by the number of samples for each class in the test set.

The shallow models had difficulty discriminating between the antagonist and mixed phenotypes and thus do not generalize well to test data. This can further be seen in Figure 4 with a comparison of confusion matrices for the three methods on the test set. The DNN predictions are noticeably better than baselines at identifying the non-responsive phenotype, and the DNN is more sensitive at distinguishing between the mixed-response and antagonist response which are subtly different.

### 2.6. Experimental Validation

We next used the trained DNN model to predict the response of 44 common mutants to darolutamide, a leading AR antagonist which has not yet shown a resistant, mixed-response phenotype. These 44 mutants had been tested with bicalutamide and enzalutamide but not with darolutamide, therefore making them an ideal validation set for our model. To make these predictions, we compute Z-scale descriptors and Morgan circular fingerprints according to the same protocol as above and then use our trained DNN model to predict the response phenotype based on these descriptors. Our model predicts that, of these 44 mutants, 12 are predicted to be non-responsive, 31 are predicted to be antagonized by the drug, and a single mutant, E666D, is predicted to have the resistant, mixed-response phenotype. Prediction results for the two baseline methods are also included; however, both methods under-perform compared to the DNN as they tend to over predict for the mixed-response phenotype.

These mutants were then tested experimentally in PC3 cell lines transfected with the corresponding AR mutants with increasing concentration of darolutamide, as described in the experimental methods section and according to a similar protocol used to produce training data. Interestingly, we see that the experimental responses of these mutants match almost exactly to the predictions made by the model, as can be seen in Figure 5, with full results in Figure 6, color-coded according to the class of response (green = antagonized, blue = non-responsive, yellow = mixed response), along with the corresponding predictions made by the DNN model (−1 = antagonized, 0 = non-responsive, 1 = mixed response). One mutant was incorrectly predicted as having an antagonist response when in fact it is non-responsive in cell experiments. Additionally, the E666D mutant which was predicted to have a resistant phenotype is instead antagonized by darolutamide in cell experiments. We, in fact, see some minor activation of this mutant and two others (L595M and T576A) at higher drug concentrations, but this activation did not exceed our cutoff of 10%. This illustrates the subtle difference between the antagonist and resistant phenotypes that is challenging to capture with the model and might also suggest that these mutants or others could show resistance at higher drug concentrations or may be resistant in combination with another mutation.

A possible explanation for the lack of resistant phenotype in response to darolutamide may be found by looking at the chemical structures of current anti-androgens (Figure 7). Bicalutamide, apalutamide, and enzalutamide have nearly the same chemical structures, sharing the same active substructure (highlighted in red) responsible for the antagonist effect on the AR, with only minor differences in the rest of the compound to change solubility and other drug properties. It is no surprise then that all three of these drugs have shown similar resistant phenotypes both in the clinic and in cell experiments. Darolutamide, the newest of the anti-androgens, has a noticeably different chemical structure in that it is a longer overall molecule with a different tail than the other drugs. However, it still shares a similar active substructure as the other molecules, although not identical. These small differences in chemical structure may be responsible for lack of resistant phenotype seen with darolutamide. However, it could also be the case that darolutamide has simply not been prescribed for as long as the other drugs, and we are yet to see resistant mutants arise in response to prolonged treatments.

## 3. Conclusions

In this work, we demonstrated the potential of deep learning in combination with flexible and informative proteochemometric descriptors to predict adverse drug responses of a wide range of acquired AR mutations observed in clinical samples and reported in the literature over the years. Indeed, the predictions made by the model for darolutamide are nearly perfectly confirmed in experimental validation and show that darolutamide appears to be less prone to resistance. However, future work will look to predict and validate many more mutants and double mutants that may be activated by darolutamide. As more data becomes available for this problem, the superior accuracy and ease of use of our DNN approach would be expected to grow. We recognize that the applicability domain of this particular model is limited to mutants of the AR and to compounds similar to the current anti-androgens. However, this modeling approach could also easily be extended to other protein targets and used to predict for other drug compounds, either experimental or in clinical use. The developed DNN model would also be of high practical utility, either in the clinical context by warning of resistance causing mutations to current therapies or in the drug development context to screen lead compounds for their likelihood to elicit resistant phenotypes.

## 4. Materials and Methods

### 4.1. AR Mutant Functional Assay

PC3 cells lacking the AR were maintained in RPMI (Roswell Park Memorial Institute)1640 media (Life Technologies, Waltham, MA, USA) and 5% FBS (Hyclone Thermo Fisher Scientific, Logan, UT, USA) at 37 ${}^{{}^{\circ}}$C and 5% CO${}_{2}$. For the functional assay, cells were seeded in 96-well plates (5000 cells/well) in RPMI 1640 medium with 5% charcoal-stripped serum (CSS) (Hyclone). After 24 h, cells were co-transfected with 25 ng of wild-type or mutated-AR and 25 ng of the reporter plasmid pARR3-tk-luciferase using TransIT 20/20 transfection reagent (3 μL/μg of DNA) (Mirus Bio LLC, Madison, WI, USA) in Optimem serum free media (Life Technologies). At 24 h after transfection, 0.1 nM R1881 and either 0.1% DMSO (Dimethyl sulfoxide, control) or serial dilutions of increasing concentrations of the studied compounds (0.0025, 0.0076, 0.023, 0.069, 0.206, 0.617, 1.852, 5.556, 16.667, and 50 μM) were added to the wells. At 24 h after treatment, the medium was aspirated off and the cells were lysed by adding 60 μL of 1× passive lysis buffer (Promega), followed by shaking at room temperature for 15 min and two freeze/thaw cycles at −80 ${}^{{}^{\circ}}$C. Twenty μL of lysate from each well was transferred onto a 96-well white flat bottom plate (Corning, Corning, NY, USA) and the luminescence signal was measured after adding 50 μL of luciferase assay reagent (Promega, Madison, WI, USA). The chemical oxidation of luciferin into oxyluciferin by the luciferase is accompanied by light production that can be quantified as luminescence by a TECAN M200Pro instrument. Each concentration was assayed in quadruplicate n = 4, with a biological replicate of n = 3. Results were averaged and normalized by expressing them as a percentage of WT (wild-type) AR activity.

### 4.2. Computational Methods

All scripts were written in python and are available along with input datasets at https://github.com/osnow/AR_mutant_response. The DNN model was implemented in Keras [34] and the baseline models implemented using skLearn [35]. Input features were calculated using rdkit [28] and the iLearn python package [24].

Raw AR drug response data were converted to the 4 classes as follows: if the average of response values across all drug concentrations was less than 5% activation, the sample was classified as non-responsive. If the response at 50 μM was lower than the response at 0 μM (decreasing trend) and less than 10% activation, the sample was classified as antagonist. If the response at 50 μM was greater than the response at 0 μM (increasing trend) and the min activation was greater than 50%, the sample was classified as agonist. If the responses showed a decreasing trend but the response at 50 μM was greater than 10% (u-shaped), the sample was classified as mixed-response. The conversion of the response curves to classes were checked against domain expert classifications for reliability.

Optimal cut-off values for RF and SVM can be seen in Table 2, which were found using Youden’s J statistic. Metrics of the performance of the models on the test set were then calculated using these optimal cut-off values.

## Figures and Tables

**Figure 1 ijms-21-05847-f001:**
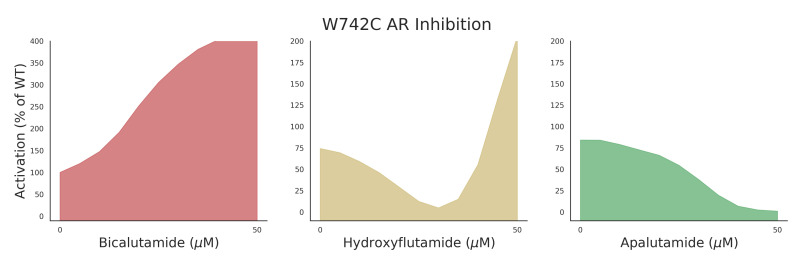
W742C androgen receptor (AR) mutant transcriptional response to increasing dose of anti-androgen. Y-axis corresponds to percent activation in luciferase reporter assay normalized to wild-type (100% indicates same activation as wild-type).

**Figure 2 ijms-21-05847-f002:**
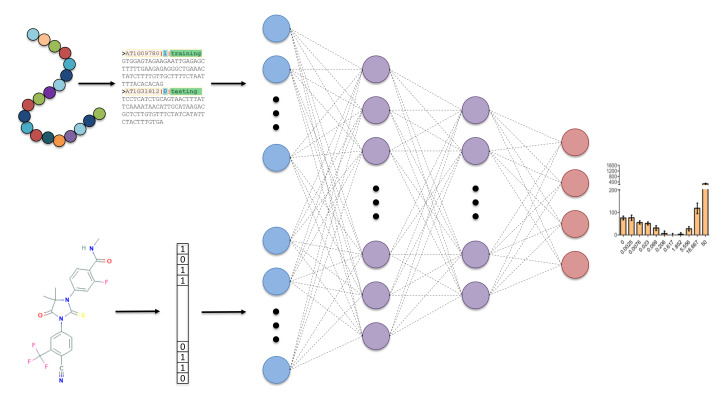
Overview of deep neural network (DNN) model taking sequence and 2D chemical descriptors as input, 2 hidden layers with 128 and 32 nodes, respectively, and a 4 node output to classify four categories of dose response curves.

**Figure 3 ijms-21-05847-f003:**
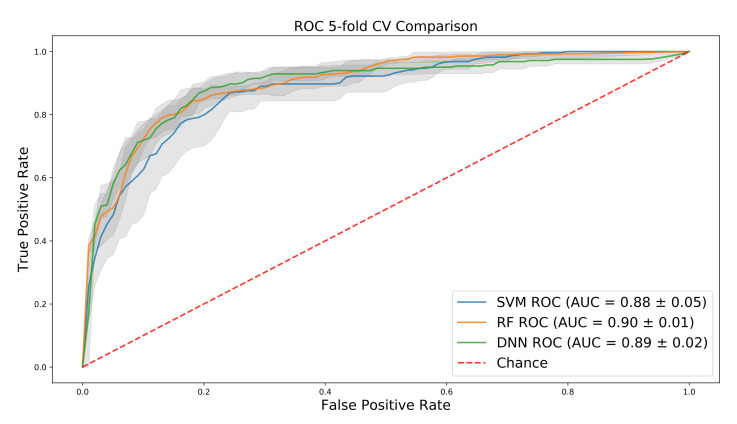
Mean ROC (Receiver operating characteristic) curves over 5-fold cross-validation for support vector machines (SVM), random forest (RF), and deep neural networks (DNN) with 1 standard deviation shown in grey.

**Figure 4 ijms-21-05847-f004:**
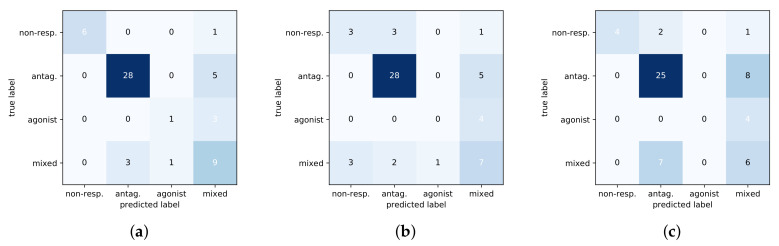
Confusion matrices of three methods on test set. Predicted classes are along the x-axis with the true class on the y-axis. (**a**) DNN. (**b**) RF. (**c**) SVM.

**Figure 5 ijms-21-05847-f005:**
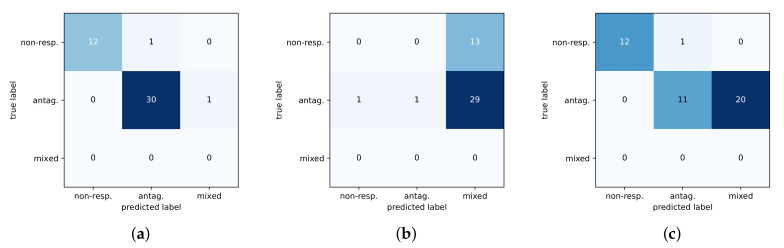
Confusion matrices for darolutamide predictions and experimental validation. Predicted classes are along the x-axis with the true class on the y-axis. The agonist class is dropped as no mutants showed this phenotype with darolutamide, and no models predicted this class. (**a**) DNN. (**b**) RF. (**c**) SVM.

**Figure 6 ijms-21-05847-f006:**
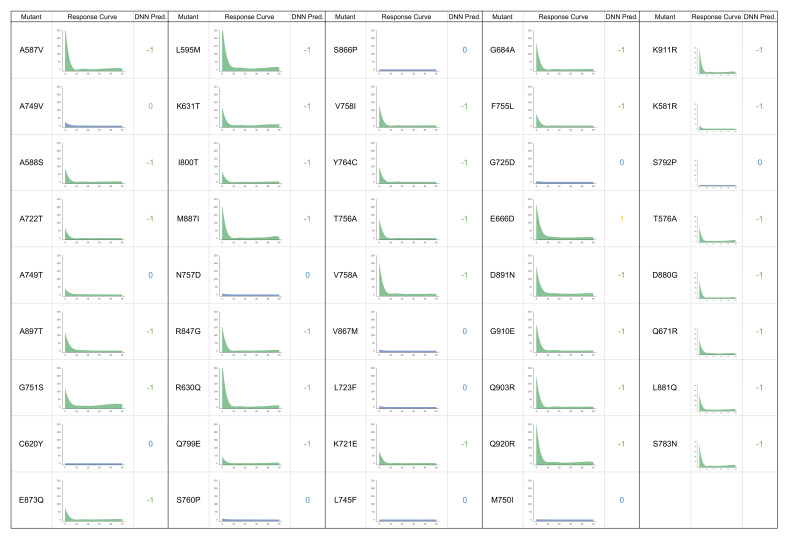
Experimental drug-response curves of AR mutants treated with darolutamide with corresponding DNN predictions. −1 = antagonized, 0 = non-responsive, and 1 = mixed-response.

**Figure 7 ijms-21-05847-f007:**
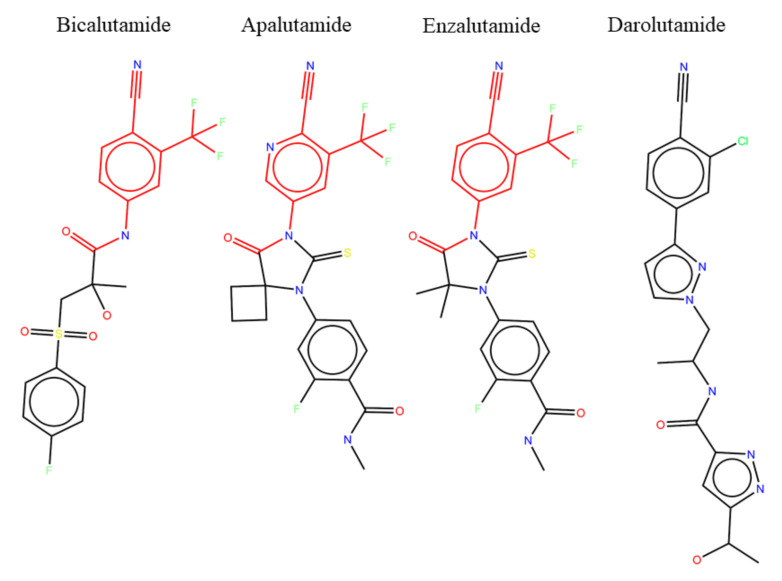
Chemical structures of anti-androgens with common substructure highlighted.

**Table 1 ijms-21-05847-t001:** Classification report containing precision, recall and F1 score for DNN and baselines on test data. 0–3 correspond to the four classes “non-responsive”, “antagonist”, “agonist”, and ”mixed-response”. Avg = weighted average for each metric according to number of samples for each class. MCC = Matthew’s correlation coefficient.

	DNN			RF			SVM		
	**P**	**R**	**F1**	**P**	**R**	**F1**	**P**	**R**	**F1**
0	1.0	0.86	0.92	0.83	0.71	0.77	1.0	0.57	0.73
1	0.9	0.88	0.89	0.93	0.87	0.90	0.70	0.72	0.71
2	0.5	0.25	0.33	0	0	0	0	0	0
3	0.56	0.71	0.62	0.5	0.71	0.59	0.32	0.43	0.35
Avg	0.80	0.79	0.79	0.75	0.75	0.75	0.62	0.61	0.6
MCC	0.654			0.601			0.336		

**Table 2 ijms-21-05847-t002:** Table of optimal cut-off values for each class for RF and SVM.

	RF	SVM
0	0.44	0.07
1	0.42	0.93
2	0.03	0.0006
3	0.2	0.05

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
