# Peer review of "Deep Learning Modeling of Androgen Receptor Responses to Prostate Cancer Therapies"

_ijms, 2020, doi:10.3390/ijms21165847_

Round 1

Reviewer 1 Report

  1. The following sentence is not clear, rewrite it. 'In this work, we leverage experimental data collected on 68 clinically observed and/or literature described AR mutants to train a deep neural network (DNN) to predict their responses to currently used and experimental anti-androgens and testosterone.' 
  2. It seems that the authors used fixed parameter settings for SVM and RF modeling whereas hyperparameter tuning used for DNN (sections 2.3 and 2.4). Ideally, the hyperparameter tuning should also be performed with SVM and RF for comparison of the performances of these machine learning methods. The authors are suggested to provide an explanation for this.
  3. What train-test ratio was used? From python script it should be 80:20 but the authors needs to mention this in the manuscript as it is a crucial information for this study. 
  4. In Table 1 it should be mentioned for what set the result is mentioned. 

Author Response

We sincerely thank the reviewer for the thorough and constructive comments which have helped us to improve the quality of the manuscript. We have incorporated all the suggestions made as they helped clarify our main thesis and strengthen our findings. The following will describe in detail the changes made and address each comment from the reviewer on a point by point basis. Changes and edits in the manuscript have been tracked in red.  

Point 1: The following sentence is not clear, rewrite it. 'In this work, we leverage experimental data collected on 68 clinically observed and/or literature described AR mutants to train a deep neural network (DNN) to predict their responses to currently used and experimental anti-androgens and testosterone.' 

Response 1: We agree this sentence was not clear and we have revised it to both read more fluidly and clarify the work done in the paper. It now reads “In this work, we leverage experimental data collected on 68 AR mutants, either observed in the clinic or described in the literature, to train a deep neural network (DNN) that predicts the response of these mutants to currently used and experimental anti-androgens and testosterone.”

Point 2: It seems that the authors used fixed parameter settings for SVM and RF modeling whereas hyperparameter tuning used for DNN (sections 2.3 and 2.4). Ideally, the hyperparameter tuning should also be performed with SVM and RF for comparison of the performances of these machine learning methods. The authors are suggested to provide an explanation for this.

Response 2: Hyperparameter tuning was in fact performed for both SVM and RF models according to the same random search cross validation protocol in order to find the best parameters. We appreciate it being pointed out that we failed to include this in the text. We have now added a sentence at lines 104-106 describing that this tuning was performed for both of these models.

Point 3: What train-test ratio was used? From python script it should be 80:20 but the authors needs to mention this in the manuscript as it is a crucial information for this study.

Response 3: The reviewer is right that we used an 80:20 split as evidenced in the python script. This information is now included in the manuscript at lines 80 and 81.

Point 4: In Table 1 it should be mentioned for what set the result is mentioned.

Response 4: The results in Table 1 correspond to the test set. The caption and the text (line 125) have been updated to clarify this.

Reviewer 2 Report

In this study, the authors constructed a machine-learning discriminant model of the dose-response curve for various mutants of the androgen receptor. This manuscript has the following important proofreading items.   1) L71-73 "Responses to six different anti-androgens (bicalutamide[18], enzalutamide [19,20, hydroxyflutamide[21], ARN509[22], and 2 in-house developed AR inhibitors)" All compound structures should be clarified in the paper.   2) L75 "The dose-response curve data consisting of 11 increasing drug concentrations and their transcriptional activity values (normalized to wild-type) were converted to 4 classes corresponding to the above phenotypes with an additional non-responsive phenotype." 2-1. "11 increasing drug concentrations" The authors should list specific concentrations in the methos section 2-2. "4 classes" The authors should describe the method of classifying the four classes of transcriptional response, which are teacher data, from experimental values   3) L78 "The dataset was split into a training set, upon which cross validation and hyperparameter tuning was performed, and a test set to evaluate the performance of the methods." 3-1. The ratio of training set to test set and the number of sample data should be described. 3-2. It is necessary to perform stratification so that the compound is uniformly contained in the split.   4) L198 "All scripts were written in python and are available along with input datasets at https://github.198 com/osnow/AR_mutant_response. " 4-1. There is almost no description of the data uploaded to github. all_muts_classes_updated.csv should add detailed explanation as supplemental data of the paper 4-2. In all_muts_classes_updated.csv, mutants and compounds should be displayed in separate columns and annotated with training/test set.   5) Table 1 5-1. MCC calculation results should also be posted. 5-2. The cutoff values of SVM and RF cannot be compared with the deep learning method unless they are optimized using the Youden Index. Authors should specify these cutoff values. 5-3. Results from both test and training data should be listed in Table 1.   6) Figure 1 6-1. The definition of "100%" should be added to the legend.   7) Figure 3 The authors shouled draw both ROC curve and precision recall curve of training set and test set, and discuss mainly using test set results.   8) Figure 5 8-1. The authors should add SVM and RF results. 8-2. Since the number of compounds is limited in this study, discussion of applicability domain is essential. The authors should explain in the text the scope of application of the compounds in this model.   9) Figure 7 Why pick up only a part of compounds.

Author Response

We sincerely thank the reviewer for the thorough and constructive comments which have helped us to improve the quality of the manuscript. We have incorporated all the suggestions made as they helped clarify our main thesis and strengthen our findings. The following will describe in detail the changes made and address each comment from the reviewer on a point by point basis. Changes and edits in the manuscript have been tracked in red.  

Point 1: L71-73 "Responses to six different anti-androgens (bicalutamide[18], enzalutamide [19,20, hydroxyflutamide[21], ARN509[22], and 2 in-house developed AR inhibitors)" All compound structures should be clarified in the paper.

Response 1: We appreciate the reviewer pointing out we did not describe the two in-house developed AR inhibitors. We have added a supplementary figure showing the chemical structures of these two AR inhibitors (VPC13566 and VPC13579) in addition to the current figure 7 that shows the structures of the other four drugs. We have also added a sentence at lines 73-74 directing the reader to this supplementary figure.

Point 2: L75 "The dose-response curve data consisting of 11 increasing drug concentrations and their transcriptional activity values (normalized to wild-type) were converted to 4 classes corresponding to the above phenotypes with an additional non-responsive phenotype." 2-1. "11 increasing drug concentrations" The authors should list specific concentrations in the methos section 2-2. "4 classes" The authors should describe the method of classifying the four classes of transcriptional response, which are teacher data, from experimental values

Response 2: 2-1. The specific concentrations are now listed in the methods section at lines 196-197 and referenced in the main text at line 77. 2-2. We added short paragraph describing how raw drug response curve data were converted to 4 classes in the methods section at lines 211-218 and referenced this section in the main text at line 79.

Point 3: L78 "The dataset was split into a training set, upon which cross validation and hyperparameter tuning was performed, and a test set to evaluate the performance of the methods." 3-1. The ratio of training set to test set and the number of sample data should be described. 3-2. It is necessary to perform stratification so that the compound is uniformly contained in the split.

Response 3: 3-1. The train test split used was 80/20 which we failed to mention in the manuscript. We thank the reviewer for pointing out this omission. This information is now included at lines 80-81. 3-2. The split was performed so that all compounds were represented in both the train and test set. We have now described this in the text at line 82. See also the response to the following point.

Point 4: L198 "All scripts were written in python and are available along with input datasets at https://github.198 com/osnow/AR_mutant_response. " 4-1. There is almost no description of the data uploaded to github. all_muts_classes_updated.csv should add detailed explanation as supplemental data of the paper 4-2. In all_muts_classes_updated.csv, mutants and compounds should be displayed in separate columns and annotated with training/test set.

Response 4: 4-1. We agree that the data was not properly described before and was in a confusing format and thank the reviewer for highlighting this. The input data has now been reformatted and uploaded in a zip file containing separate train and test files for both features and labels with the features files being a concatenation of the fingerprints and Z-scale descriptors. 4-2. We agree it would be helpful to view the responses as a matrix of mutants by drugs. Thus we have created and uploaded an additional file called “mutantsxdrugs.csv” that has all the mutants as rows and the drugs as columns with their response class label as the values.

Point 5: Table 1 5-1. MCC calculation results should also be posted. 5-2. The cutoff values of SVM and RF cannot be compared with the deep learning method unless they are optimized using the Youden Index. Authors should specify these cutoff values. 5-3. Results from both test and training data should be listed in Table 1.

Response 5: 5-1. We have now calculated and added MCC results to Table 1. 5-2. We appreciate the reviewers suggestion of optimizing cut-off values using the Youden Index. This was not a metric were very familiar with but after doing some reading and experimenting we now understand the value of the metric and its use in comparing algorithms. We believe it makes logical sense in our case to apply this metric and indeed we saw some improvements in the predictions of the baseline methods after doing this optimization. Values in Table 1 have been updated to reflect these changes and we also add a table in section 4.2 that specifies these cut-off values for each class and method. We also add two sentences at lines 219-221 describing these cutoffs. 5-3. We understand the reviewers desire to see training data results as well and thus we have calculated results for all models on the training data. However, we opt to include them as supplementary rather than include them in table 1 because all three models tend to fit the training data quite well and result in strong performance in all metrics. We believe the training results do not add much to the point made in table 1 and only serve to make it more confusing and contain too much information. Training data results are contained in a supplementary csv file.

Point 6: Figure 1 6-1. The definition of "100%" should be added to the legend.

Response 6: A sentence has been added to the caption of figure 1 describing the y-axis and the definition of 100% as it relates to wild-type activation.

Point 7: Figure 3 The authors shouled draw both ROC curve and precision recall curve of training set and test set, and discuss mainly using test set results.

Response 7: The intent of the ROC curve was to show the average performance of each model on the training set across 5-fold cross validation. The conclusion we draw from this is that each model performs relatively well in terms of AUC and there are not large differences between the models. The main focus, as the reviewer states, is on the test data, which the models have not seen before and these results are contained in table 1. We add an emphasis in the text and the caption of the table that these results are for the test data set. As per the reviewer’s suggestion, we have produced precision recall curves for each algorithm on the test set and included them as supplementary. We believe the same information is contained in a more succinct format in table 1. We feel that it is easier to parse the values in table 1 than the precision recall curves, which tend to be noisy due to the relatively small size of the dataset and also add significantly to the length of the manuscript.  

Point 8: Figure 5 8-1. The authors should add SVM and RF results. 8-2. Since the number of compounds is limited in this study, discussion of applicability domain is essential. The authors should explain in the text the scope of application of the compounds in this model.

Response 8: 8-1. Confusion matrices for SVM and RF have now been added to figure 5 as per the reviewer’s suggestion. We also add a sentence at lines 142-143 describing these results. 8-2. We agree with the review that the number of compounds in this study is limited and thus the scope of application is also limited. We admit that this particular model will only be applicable for mutations in the AR and for anti-androgen like drugs and we take care not to oversell its predictive abilities outside of that. However, we do feel that our reframing of the QSAR problem to predict the response of a range of mutations to a small panel of drugs as well as the use of a flexible DNN based approach with protein sequence features and fingerprints is a blueprint that could be applied to many other protein targets and especially other nuclear receptors which have been shown to have a similar acquired resistance. We have added two sentences in the conclusion (lines 180-183) emphasizing the limits of this particular model but also the utility of this type of approach for both future drug development and clinical screening.

Point 9: Figure 7 Why pick up only a part of compounds.

Response 9: The intent of this figure was to highlight the common substructure of the 3 older antiandrogens which share a highly similar ‘warhead’ targeting the androgen receptor, whereas darolutamide (the newer antiandrogen) has a much different and larger structure, potentially meaning it is less prone to required resistance which we find in our predictions and validation.

Round 2

Reviewer 2 Report

The authors responded almost appropriately to my comments. However, acceptance of the paper is not recommended until the following problems will be resolved.   1) The compound names must be unified in the P3L71 text, Figure 7, Figure S1, and the "mutantsxdrugs.csv" file registered on Github.   2) Seven types of compound names are written in the csv file on github. This contradicts the six types of compound information described in the paper. The authors need an explanation for this contradiction in their manuscript.

Author Response

Response to Reviewer 2 (round 2)

We are pleased that our previous revisions were satisfactory and that there are no major outstanding issues with the manuscript. We have fixed the two following discrepancies with compound naming that the reviewer astutely pointed out and changes are highlighted in red in the manuscript text.

Point 1: The compound names must be unified in the P3L71 text, Figure 7, Figure S1, and the "mutantsxdrugs.csv" file registered on Github

Response 1: We appreciate the reviewer's thoroughness in pointing out the inconsistency in the naming of the compounds. We have updated the "mutantsxdrugs.csv" file in Github to have full compound names in the header, which now match the names in Figure 7 and Figure S1. The text at lines 72-73 has also been changed so that all compounds are listed with their full name and are consistent with the figures and supplemental file. ARN509 is now Apalutamide and VPC-13566 and VPC-13789 are now listed in the tested compounds.

Point 2: Seven types of compound names are written in the csv file on github. This contradicts the six types of compound information described in the paper. The authors need an explanation for this contradiction in their manuscript.

Response 2: We again thank the reviewer for noticing this omission of ours. The text should, in fact, include seven types of compounds as some mutants were also tested against darolutamide (this compound was described later in the manuscript but should be described at the beginning as well). We have now updated the text at line 71 from 'six' compounds to 'seven' compounds and also included 'darolutamide' at line 73 with the appropriate reference.